# Polycystic Ovary Syndrome and the Potential for Nanomaterial-Based Drug Delivery in Therapy of This Disease

**DOI:** 10.3390/pharmaceutics16121556

**Published:** 2024-12-04

**Authors:** Mingqin Shi, Xinyao Li, Liwei Xing, Zhenmin Li, Sitong Zhou, Zihui Wang, Xuelian Zou, Yuqing She, Rong Zhao, Dongdong Qin

**Affiliations:** 1First Clinical Medical College, Yunnan University of Chinese Medicine, Kunming 650500, China; shimingqin1998@163.com (M.S.); ynutcmxingliwei@163.com (L.X.); 2School of Basic Medical Sciences, Yunnan University of Chinese Medicine, Kunming 650500, China; lxy9081720@163.com (X.L.); 18725261705@163.com (S.Z.); 15887447630@163.com (X.Z.); syq1731671485@163.com (Y.S.); 3School of Traditional Chinese Medicine, Changchun University of Chinese Medicine, Changchun 130021, China; lizhenmin1996@163.com; 4Second Clinical Medical College, Yunnan University of Chinese Medicine, Kunming 650500, China; 13187806753@163.com

**Keywords:** polycystic ovary syndrome, nanoparticles, novel drug delivery systems, therapeutic efficacy

## Abstract

Polycystic ovary syndrome (PCOS) is the predominant endocrine disorder among women of reproductive age and represents the leading cause of anovulatory infertility, which imposes a considerable health and economic burden. Currently, medications used to treat PCOS can lead to certain adverse reactions, such as affecting fertility and increasing the risk of venous thrombosis. Drug delivery systems utilizing nanomaterials, characterized by prolonged half-life, precision-targeted delivery, enhanced bioavailability, and reduced toxicity, are currently being employed in the management of PCOS. This innovative approach is gaining traction as a favored strategy for augmenting the therapeutic efficacy of medications. Consequently, this paper discusses the roles of nanoparticles, nanocarriers, and targeted ligands within nanomaterial-based drug delivery systems, aiming to identify optimal methodologies for treating PCOS using nanomaterials. Additionally, prospective research avenues concerning nanomaterial-based delivery systems in the context of PCOS, as well as the implications of existing insights on the advancement of novel therapies for PCOS, are highlighted.

## 1. Introduction

Polycystic ovary syndrome (PCOS) is a multifaceted condition identified by elevated androgen levels, menstrual irregularities, the presence of polycystic ovaries, and related metabolic and psychological consequences. PCOS is the most common endocrine condition impacting women of reproductive age, representing the leading cause of anovulatory infertility and a noteworthy factor contributing to the early development of type 2 diabetes mellitus (T2DM) and psychiatric issues [1]. Furthermore, demonstrate a greater incidence of pregnancy-related problems, such as early miscarriage, gestational diabetes, hypertension, and preeclampsia, along with a markedly elevated risk for endometrial cancer [2,3,4]. Research indicates that approximately 11–13% of women worldwide are affected by PCOS, leading to considerable healthcare and economic challenges [5,6,7]. Current effective management strategies for PCOS primarily focus on addressing metabolic function, reproductive health, hyperandrogenism, and psychological well-being. Common pharmacological treatments include metformin, clomiphene citrate, ethinylestradiol, and cyproterone acetate [8,9]. Nonetheless, these medications may carry potential adverse effects, such as hindering conception and raising the risk of venous thromboembolism [5].

In recent years, drug delivery methods utilizing nanomaterials have garnered significant attention due to their unique advantages, including nanoscale dimensions, controlled release of therapeutic agents, superior biocompatibility, remarkable optical and physical characteristics, and straightforward surface modification [10]. Recent findings indicate that nanomaterials can effectively transport conventional therapeutics to mitigate damage caused by inflammatory responses and address endocrine dysfunction, thereby reversing the pathological alterations associated with the progression of PCOS [11]. Hence, nanomaterials present substantial potential for enhancing the treatment of PCOS. This paper provides a comprehensive overview of the latest advancements in nanocarrier delivery strategies, targeting molecules, and drug release triggers, aiming to identify the most effective combinations of nanomaterial-based delivery systems for PCOS management. Furthermore, it will also explore future research directions pertaining to nanodelivery strategies (Figure 1).

## 2. The Pathophysiology of PCOS

High androgen levels, irregular menstrual cycles, and the presence of polycystic ovaries are the hallmarks of PCOS, a multifaceted disorder. The pathophysiological mechanisms underlying PCOS are complex and include dysfunctions within the hypothalamic–pituitary–ovarian (HPO) axis, influenced by hyperandrogenemia, insulin resistance (IR), impaired steroidogenesis, and adipose tissue accumulation [12]. The hypothalamic secretion of gonadotropin-releasing hormone (GnRH) in the HPO axis is essential for the regulation of both androgen production and insulin resistance. A disruption in GnRH secretion can result in decreased levels of follicle-stimulating hormone (FSH) and increased levels of luteinizing hormone (LH) [13]. Previous studies have indicated that insulin, in conjunction with *LH*, can enhance the generation of androgens within follicular membrane cells [14,15,16]. Moreover, excessive secretion of adrenocorticotropic hormone (ACTH) can lead to an overproduction of androgens by the adrenal cortex [17]. This state of hyperandrogenemia disrupts follicular maturation and ultimately hinders ovulation.

The underlying pathophysiological mechanisms of PCOS remain inadequately explored. Current research indicates that hyperandrogenemia, IR, and hyperinsulinemia serve as primary contributors to female reproductive dysfunction [18,19,20,21,22,23]. Furthermore, inflammatory responses and oxidative stress can adversely influence oocyte quality and endothelial function, thereby facilitating the onset of PCOS [18].

### 2.1. Hyperandrogenism

Hyperandrogenemia presents as a systemic condition characterized by elevated androgen levels. In individuals diagnosed with PCOS, this condition may stem from impaired endogenous steroidogenesis in ovarian cells [19]. The androgens circulating in the bloodstream of women include dehydroepiandrosterone sulfate (DHEAS), testosterone (T), dihydrotestosterone, dehydroepiandrosterone (DHEA), and androstenedione (A) [20]. In the context of PCOS, there is an overproduction of *DHEAS*, *T*, *DHEA*, and *A*, which contributes to premature development of ovarian follicles, formation of numerous small antral follicles, and anovulation [14]. Additionally, heightened peripheral cortisol metabolism is posited as another contributor to hyperandrogenemia. Reduced cortisol levels disrupt negative feedback in the hypothalamic–pituitary–adrenal (HPA) axis, leading to an increase in the synthesis of *ACTH* by the pituitary gland, and this stimulates adrenal steroidogenesis [21]. Therefore, various pathways can lead to excessive androgen levels, resulting in hyperandrogenemia, which is one of the primary underlying causes of the pathophysiology of PCOS. Furthermore, hyperandrogenemia is posited to be one of the potential contributors to IR in PCOS patients. Research by Paolo et al. [22] indicated that individuals with a hyperandrogenic phenotype exhibited characteristics of insulin resistance. The administration of antiandrogenic pharmacological agents has been suggested to ameliorate insulin resistance in this population [23].

### 2.2. IR

Insulin serves diverse functions across multiple tissues to maintain equilibrium between nutrient availability and requirement. Under normal physiological circumstances, rising blood glucose levels trigger heightened insulin secretion, facilitating glucose uptake by peripheral tissues while simultaneously suppressing hepatic gluconeogenesis. When the capacity of insulin to fulfill these metabolic functions is compromised, compensatory hyperinsulinemia emerges, a phenomenon commonly referred to as insulin IR [24]. IR is notably prevalent among individuals with PCOS, with estimates suggesting that around 75% of PCOS patients develop IR [25]. This condition is intricately linked to metabolic disturbances associated with PCOS. Research from the 1990s posited that insulin engages its specific receptor in PCOS, thereby promoting not only ovarian and adrenal steroidogenesis but also the secretion of LH from the pituitary organ [26]. The hyperinsulinemia induced by tissue IR is fundamental to the pathophysiology of PCOS [27]. Furthermore, IR in PCOS patients impacts metabolic or mitogenic pathways in non-traditional insulin-responsive tissues such as the ovaries and the pituitary gland [28]. Additionally, the HPO axis is disrupted, and the progression of PCOS is accelerated, as IR intensifies hyperandrogenemia, according to compelling evidence [28].

### 2.3. Inflammatory Response and Oxidative Stress

According to research, patients diagnosed with PCOS demonstrate elevated levels of genetic and inflammatory markers. Investigations have demonstrated a substantial correlation between elevated concentrations of C-reactive protein (CRP), interleukin-18 (IL-18), tumor necrosis factor-alpha (TNF-α), interleukin-6 (IL-6), and ferritin in women with PCOS and control groups that are age- and BMI-matched [29,30,31,32]. Within the context of PCOS, hyperandrogenemia incites an inflammatory response alongside IR, both of which amplify reactive oxygen species (ROS) production and subsequently facilitate oxidative stress [33].

Studies have shown that oxidative stress disrupts normal follicular development and maturation [34,35]. An excess of ROS can harm both oocytes and granulosa cells in the follicle, compromising their integrity and posing a risk to fertility [36,37]. Oxidative stress is notably linked to pathological conditions associated with PCOS, including IR, hyperandrogenism, inflammatory responses, and obesity. Frank González et al. [38] have proposed that hyperglycemia may elevate ROS levels, resulting in a pro-inflammatory environment that triggers IR and hyperandrogenism in individuals with PCOS. Furthermore, insulin resistance may be facilitated by compromised oxidative phosphorylation and mitochondrial dysfunction, which disrupt insulin signaling pathways and impede glucose metabolism [39,40]. In obese individuals, markers of antioxidant activity such as superoxide dismutase (SOD) and glutathione peroxidase (GSH-Px) are significantly diminished [41].

## 3. Diagnosis and Treatment of PCOS

### 3.1. Diagnosis of PCOS

In 1935, Stein et al. [42] first recognized what is currently termed PCOS in their study. Since that time, various diagnostic criteria for PCOS have emerged, sparking a contentious debate that persisted until 2003. During this year, 27 PCOS experts convened in Rotterdam to establish a consensus statement referred to as the “Rotterdam Criteria” [43]. These criteria expand the phenotypic characterization of PCOS to encompass any two of the three primary features: oligomenorrhea, hyperandrogenism, and a polycystic ovarian pattern observed via ultrasound. The scientific investigation of PCOS has been impeded by the coexistence of multiple classification systems, which has led to clinical ambiguity. Consequently, in 2012, experts advocated for the adoption of the broader 2003 Rotterdam Criteria while also delineating subphenotypes within these criteria. Nevertheless, the Rotterdam criteria continue to be the predominant choice among clinicians across various specialties [44].

Two of the following three manifestations—clinical or biochemical hyperandrogenemia, oligomenorrhea or amenorrhea, and polycystic ovaries—must be identified in order to diagnose PCOS using the Rotterdam criteria [45]. Clinical manifestations of hyperandrogenism include acne, hirsutism, and androgenetic alopecia, while biochemical indicators may indicate elevated levels of total, free, or bioavailable testosterone [46]. Oligomenorrhea is classified as menstrual intervals exceeding 35 days or experiencing 5–9 menstrual cycles annually. Amenorrhea refers to the cessation of menstruation for a period of three months or more [47]. A polycystic ovarian morphology is defined as the presence of 20 or more follicles per ovary, with a diameter spanning from 2 to 9 mm, and/or an ovarian volume exceeding 10 mL, as determined by ultrasound [42]. Additionally, the diagnosis of PCOS is exclusive, mandating that healthcare practitioners exclude other prevalent conditions presenting with similar clinical, biochemical, and morphologic features. These conditions encompass thyroid disorders, hyperprolactinemia, and non-classic congenital adrenal hyperplasia [48]. For instance, individuals with primary hypothyroidism may experience menstrual irregularities, infertility, and weight gain [49]. Laboratory evaluations indicate minor increases in free and total testosterone, free and total estradiol, prolactin, and *LH* [49]. Changes in ovarian morphology, such as a bilateral polycystic appearance, may also be evident on ultrasound [49].

### 3.2. Treatment of PCOS

The intricate pathogenesis and characterization of PCOS necessitate a multifaceted treatment approach. This aims to alleviate diagnosed symptoms and support patient well-being. An international survey identified obesity, menstrual irregularities, infertility, and excessive hair growth as critical health concerns for PCOS patients [50]. Consequently, therapeutic strategies should prioritize these issues. The 2023 International Evidence-Based Guidelines for the Evaluation and Management of PCOS designate oral contraceptives as the primary treatment option for addressing menstrual irregularities and androgen excess [5]. Current medications mainly target specific symptoms linked to PCOS, with no drugs explicitly approved for the syndrome itself. Effective treatments should address health risks, mitigate key pathophysiological processes, and cater to individual symptoms and needs. When applicable, interventions should aim to restore ovulatory cycles, enhance fertility, normalize menstrual cycle length, reduce clinical and biochemical hyperandrogenemia, improve insulin sensitivity, promote weight reduction, lower cardiometabolic risks, and enhance disease-specific quality of life.

Current pharmacological options for PCOS primarily include contraceptives, anti-androgens, and metformin. However, these treatments carry potential side effects. For instance, combined contraceptives with androgenic progestins like norethindrone risk worsening hyperandrogenic symptoms [51]. Anti-androgenic drugs may lead to feminization of male fetuses [5]. Additionally, insulin sensitizers are predominantly indicated for T2DM but are commonly utilized for PCOS [52]. Adverse effects can manifest as treatment resistance, low adherence, or even treatment discontinuation. Therefore, there is a pressing need to improve the efficacy of the current therapeutic agents for PCOS and to investigate strategies to alleviate their associated side effects.

## 4. Nanomaterials in PCOS Treatment

Over the last thirty years, nanomedicines have progressed significantly in pharmacology. They have significantly enhanced the utility of some medicines [53,54]. Numerous effective nanotherapies emerged for various human diseases [55,56,57,58]. Recently, advancements in nanotechnology have been noted in treating PCOS. The key mechanisms by which PCOS leads to anovulatory infertility and metabolic dysfunction involve multiple aspects, including hyperandrogenism, insulin resistance, inflammatory response, and oxidative stress. Nanomaterial-based therapy can intervene in these processes through various pathways. Nanocarriers serve as vehicles to transport drugs targeting these diseases. Presently, we utilize different nanocarriers for PCOS-related conditions. These include nanoparticles, liposomes, carbon nanotubes, quantum dots, and micelles. Their benefits, challenges associated with PCOS, and the methods for drug administration are outlined below (Table 1).

### 4.1. Nanoparticle

#### 4.1.1. Natural-Based Drug Nanoparticles

Natural medicines encompass a range of therapies derived from animal, plant, and mineral sources, all of which modern pharmacological research has validated for specific bioactive properties. Recent trends indicate a notable rise in plant-derived drugs due to their potential rehabilitative, therapeutic, and preventative roles in herbal medicine [70,71]. Arentz et al. [72] revealed dissatisfaction with Western medication among women with PCOS and their supplemental medication utilization rate of more than 70%, which suggests patient preference for complementary therapies. In addition, traditional medications associated with the treatment of PCOS disorders have side effects and are likely to be ineffective in some cases [73,74]. Various herbs have recently garnered attention for their applications in PCOS, providing multiple active compounds that may yield synergistic effects [75]. A substantial number of botanicals are known to contain pharmacologically active constituents that can have a beneficial impact on ovulatory function, obesity, insulin sensitivity, and hormonal levels, according to previous research [76]. For example, Maharjan et al. [77] demonstrated that Aloe barbadensis gel positively impacted PCOS in an animal model induced by letrozole, while chamomile extract facilitated normal follicular development in PCOS-affected rats [78]. Javad Heshmati et al.’s clinical trial suggested that curcumin might serve as a safe and effective adjunct therapy for alleviating hyperandrogenism and hyperglycemia linked to PCOS [79].

In recent years, the therapeutic applications of natural drug nanoparticles have emerged, with properties like surface charge and size distribution paralleling those of mammalian extracellular microbubbles [80,81]. These nanoparticles act as natural nanocarriers for vesicle transport and possess structural and functional biomolecules that enhance their clinical utility [82]. This positions such nanovesicles as viable candidates for developing next-generation biotherapeutics and drug delivery systems, addressing pressing clinical needs.

Curcumin, a phenolic compound that is acknowledged for its anti-inflammatory and antioxidant properties, has demonstrated efficacy in reducing hyperandrogenism, insulin resistance, and hyperglycemia in PCOS patients [83,84]. However, its clinical utility is limited by solubility issues and suboptimal pH stability. Researchers are increasingly focusing on biocompatible polymeric nanoparticles, like chitosan (CS), which are readily manageable and dissolve quickly [54,85,86]. While CS exhibits low water solubility, this limitation can be mitigated through chemical modifications [87,88,89]. Functionalizing CS allows for the encapsulation of various drug molecules, enhancing the pharmacokinetic profiles of these active substances [90]. As a cationic polysaccharide, CS nanoparticles are advantageous for embedding diverse compounds, including antimicrobials, analgesics, and anti-inflammatory agents [91]. Raja et al. [59] successfully developed nanoparticles featuring curcumin-embedded arginine and N-acetylhistidine-modified CS, effectively inhibiting serum LH, prolactin, testosterone, and insulin levels in a rat model compared to controls. This research marks an encouraging initial stride toward leveraging nanoparticles as a potent delivery mechanism for curcumin in treating PCOS.

Ginger originates from the rhizome of Zingiber officinale, containing a variety of bioactive compounds, including phenols, terpenes, and volatile oils. The bioactive component, 6-gingerol, imparts a robust flavor while lowering insulin levels and enhancing insulin sensitivity [92]. Anil Kumar et al. [60] revealed that ginger-derived nanoparticles also combat insulin resistance, possibly by enhancing Foxa2 expression, thus mitigating IEC exosome-mediated insulin resistance.

Cinnamon belongs to a diverse genus utilized for centuries in culinary, spice, and medicinal contexts. All cinnamon species contain various minerals such as cinnamaldehyde, eugenol, manganese, iron, calcium, dietary fiber, and related compounds [93]. Clinical and animal trials have examined cinnamon’s therapeutic benefits for PCOS [94]. Evidence indicates that cinnamon restores the estrous cycle, lowers testosterone and insulin levels, and elevates LH levels in murine models of PCOS [95]. A clinical study by Borzoei et al. involving 84 overweight women with PCOS demonstrated that brief cinnamon supplementation positively impacts metabolic risk factors [96]. Koffi Kouame et al. [61] identified Cinnamomum cassia-derived silver nanoparticles (CcAgNPS) potentially enhancing renal function in diabetic rats through their antioxidant properties.

Moreover, nanoparticles sourced from aloe vera [97], camellia [98], and other natural extracts demonstrate antioxidant and anti-inflammatory effects, possibly emerging as novel therapeutic targets for future PCOS management.

#### 4.1.2. Metal Nanoparticles

Traditionally, researchers synthesized nanoparticles through physical and chemical means, which present several challenges. Recent studies have shifted focus toward innovative therapeutic nanoparticles like silver and selenium for PCOS treatment [99]. Silver nanoparticles show promise in treating inflammation and reducing inflammatory markers in PCOS-afflicted rats [62]. Such advancements signal a progressive approach, as alleviating inflammation could diminish symptoms often associated with elevated inflammatory cytokines in PCOS patients.

Current investigations predominantly utilize metal nanoparticles via natural compounds to maximize therapeutic benefits. Silver nanoparticles demonstrate impressive antibacterial properties, exhibiting activity against cinnamon extracts noted for anticancer and anti-inflammatory effects [61,100,101]. Research indicates that curcumin-loaded iron nanoparticles can effectively inhibit apoptosis in ovarian injury cells, thereby providing potential benefits for PCOS management [63]. Selenium nanoparticles have also shown effectiveness in PCOS treatment by reducing androgen synthesis and disrupting the excessive androgen cycle via diminished androgen receptor expression [64]. Additionally, selenium nanoparticles have been found to attenuate insulin sensitivity markers, sex hormone concentrations, inflammation, and mitochondrial function in PCOS models [65]. Furthermore, research indicates that carbon nanoparticles selectively enhance follicle-stimulating hormone (FSH) levels, while copper nanoparticles improve ovarian cell viability and steroid secretion [40].

Natural nanoparticles and metal nanoparticles present distinct characteristics regarding biocompatibility, immune response potential, and degradation kinetics. Natural nanoparticles generally exhibit favorable biocompatibility, typically eliciting minimal immune reactions within biological systems, and they tend to biodegrade relatively quickly due to their resemblance to biological tissues [102,103]. Conversely, the biocompatibility and degradation rates of metal nanoparticles are contingent upon the specific metal used and any surface modifications applied. Surface-modified metal nanoparticles can function effectively as vaccine delivery systems, triggering a significant immune response [104,105,106]. Consequently, in the selection of nanoparticles, their biocompatibility, potential immune response, and degradation rate should be comprehensively considered.

### 4.2. Liposome

Liposomes represent self-assembled, spherical systems featuring lipid bilayers composed of one or more phospholipids [107]. These structures are both non-toxic and biodegradable, providing a versatile delivery mechanism for various pharmaceuticals [108]. They are capable of transporting a variety of substances, such as proteins, nucleic acids, and small molecules, which are hydrophobic and hydrophilic. By stabilizing compounds, liposomes enhance the therapeutic efficacy of drugs, overcome cellular uptake barriers, and facilitate the bio-distribution of drugs to target sites while mitigating systemic toxicity [109]. Resveratrol, a natural polyphenol, can enhance reproductive health, yet its application is restricted by low bioavailability. DMU-212, a methoxy derivative of resveratrol, demonstrates enhanced bioavailability as a result of its lipophilic properties. Lipid nanoparticles (LNPs) significantly enhance the bioavailability of drugs with limited permeability by facilitating their passage through gastrointestinal barriers, improving transmembrane absorption, and optimizing transport kinetics. Current challenges and future opportunities for oral drug delivery systems involve the integration of computational tools [110]. Nanoparticle drug delivery systems (NDDSs) play a crucial role in enhancing oral bioavailability by promoting intracellular penetration and augmenting drug absorption. On one hand, oral drug delivery faces biochemical and physiological obstacles, while on the other hand, the characteristics of the nanoparticles—such as size, surface properties, and morphology—are critical factors influencing their effectiveness [111]. Researchers developed a liposomal version of DMU-212 (lipDMU-212) and demonstrated that lipDMU-212 substantially increases the secretion of estradiol and progesterone in a dose-dependent manner [66]. This finding indicates potential benefits of lipDMU-212 in treating conditions linked to estrogen deficiency and hyperandrogenism, such as PCOS.

### 4.3. Nanotubes

Carbon nanotubes are a new generation of materials that are extensively employed in biomedical applications due to their unique structure, attractive properties (e.g., size and aspect ratio of covering surface area to length), and electrical, mechanical, and thermal properties [112]. Compared to other nanocarriers, carbon nanotubes can be easily modified to bind bioactive compounds and ligands for targeting. Therefore, the application of carbon nanotube-based nanomedicine is very attractive and promising. Miey Park et al. [67] reported that carbon nanotubes can exert a role in insulin resistance by reducing fasting blood glucose and improving serum biomarker levels associated with oxidative stress and inflammation. Carbon nanotubes are employed for diagnostic applications in addition to drug delivery. Human fetuin A (HFA) is a biomarker associated with PCOS insulin resistance as well as the inflammatory process of T2DM. Esther Sánchez-Tirado et al. [113] used magnetic multi-walled carbon nanotubes (m-MWCNTs) as nanocarriers to measure HFA, and the results showed that the performance of this assay was significantly superior to that reported by ELISA kits and chronoimpedance immunosensors. The study by Pradeep K. Jha et al. [114] found that multi-walled carbon nanotubes are capable of loading silver nanoparticles and may be used in the diagnosis of infertility in the future. Furthermore, the toxicity of carbon nanotubes is an issue that must not be disregarded. Jianbin Zhao et al. [115] discovered that prolonged exposure to multi-walled carbon nanotubes (MWCNTs) adversely impacts the growth and development of X. tropicalis and presents a possible reproductive concern. At the same time, MWCNTs in humans may also inhibit progesterone secretion from ovarian granulosa cells, possibly by inhibiting steroid acute regulatory protein expression [116].

### 4.4. Quantum Dots

Quantum dots exhibit remarkable photostability and high brightness, significantly surpassing other fluorescent dyes. They find application in various immunoassays involving antibodies. Kunal Sarkar et al. [68] reported a pegylated graphene oxide quantum dot (GOQD)-based nanodrug delivery platform. This platform facilitates the sustained release of metformin, potentially because GOQD-PEG can restore glucose uptake and alleviate insulin resistance in an in vitro model of insulin resistance. Additionally, quantum dots can integrate with nanoparticles, producing multisensing effects [117]. Generally, in mammalian systems, quantum dots penetrate cell membranes and regulate substance transport across the cell barrier. This largely transpires via mechanisms like clathrin-mediated endocytosis, caveolae-mediated endocytosis, and micropinocytosis [118,119,120]. Recent findings reveal that quantum dots can sensitively detect and localize unique markers for ovarian cancer metastasis, enabling precise treatment strategies [121]. A study by Vimal Singh et al. [122] demonstrated the safety and non-toxicity of carbon quantum dots in both mice and cervical cells, supporting their viability for biomedical applications.

### 4.5. Micelles

Micelles, extensively used molecular assemblies, have drawn considerable attention in contemporary medicine due to their remarkable stability. Polymeric micelles demonstrate reduced drug immunogenicity and prolonged half-life compared to traditional micelles [123]. Co-delivery micelles may enhance chemotherapy efficacy while somewhat mitigating reproductive toxicity and fertility damage [124]. Micellar nanoparticles arise from the self-assembly of surfactants or amphiphilic polymers in aqueous conditions, functioning as targeted drug delivery vehicles. Research by Maryam Nazarian et al. [69] indicated that curcumin and nano-membicellated curcumin (NMC) effectively reduce chlorpyrifos-induced reproductive toxicity, likely by lessening oxidative damage and inflammatory responses. Interestingly, another study indicated that NMC might diminish oocyte quality and impact embryonic development [125]. Notably, a newly developed siRNA nanoparticle, self-assembled micellar inhibitory RNA (SAMiRNA), shows promise in treating androgenetic alopecia linked to hyperandrogenism [126]. Oral insulin, a potential treatment for PCOS patients, encounters low bioavailability due to enzyme instability and malabsorption. Micelles offer great potential as a novel drug delivery method for oral insulin, demonstrating greater sensitivity to glucose levels and rapid responses to glucose fluctuations [127].

### 4.6. Nanoparticle Toxicity Profiles and Relevant Safety Measures

While nanoparticles offer a multitude of benefits in drug delivery systems, their potential toxicity must not be disregarded. These particles have the ability to traverse membrane barriers and disseminate throughout the body via the circulatory system, impacting various organs and tissues [128]. The interaction between nanoparticles and cellular structures can result in nanotoxicity. Similar to pharmaceuticals or chemical agents, the toxicity of nanoparticles is dependent on the route of administration and the extent of environmental exposure [129]. In biological contexts, the toxicological profiles of nanoparticles are primarily influenced by mechanisms such as the production of ROS, disruption of membrane integrity, and DNA damage. Consequently, a range of strategies should be employed to mitigate these risks. For example, applying a layer of biocompatible material to the surface of nanoparticles can significantly diminish their toxicity [130]. Additionally, altering the surface properties of nanoparticles represents a crucial strategy for toxicity reduction. For instance, functionalizing the surface of iron oxide nanoparticles with organic compounds can stabilize their inherent chemical reactivity and improve biocompatibility, thereby lessening toxicity [131].

## 5. Targeted Ligands in Nanomaterial-Based Drug Delivery Systems for PCOS

Nanomaterials can enhance the solubility and bioavailability of pharmaceuticals [132]. By encapsulating drugs within nanoparticles, it is possible to protect them from enzymatic degradation in the body, thereby improving their stability and bioavailability. Nanomaterials can alter the pharmacokinetic properties of drugs, including absorption, distribution, metabolism, and excretion [133]. Nanoparticles can prolong the circulation time of drugs in the body and reduce their clearance rate, which enhances efficacy and minimizes side effects [134]. Targeted drug delivery can be achieved by modifying the surface of nanoparticles with specific ligands. This targeting mechanism can increase the concentration of drugs at pathological sites while reducing their impact on normal tissues, thereby improving therapeutic outcomes and reducing adverse effects. The small size of nanoparticles allows for easier penetration through biological barriers, such as cell membranes and tissues, which increases drug concentration at the target site. Additionally, nanoparticles can diminish the systemic distribution of drugs, thereby reducing side effects on non-target organs [135]. Research indicates that the encapsulation of metformin within nanoparticles can prolong its half-life and enhance the drug’s bioavailability [136]. Clomiphene citrate (CC) is a widely utilized medication for ovulation induction; however, it may adversely affect endometrial receptivity. A study conducted by Marziyeh Ajdary and colleagues reveals that a novel nanoparticle drug delivery system, specifically a Phosal-based formulation (PBF) containing CC, can optimize the targeted delivery of CC while alleviating its detrimental effects on endometrial receptivity [137]. Additionally, nanoparticles can facilitate an increase in the dissolution rate and absorption efficiency of the drug within the gastrointestinal tract.

Certain drug delivery carriers mentioned above lack targeting capabilities; thus, it is essential to engineer targeting ligands on the surfaces of these nanocarriers to enhance the nanoparticles’ targeting qualities. These targeting ligands can be mainly classified into two categories based on the different receptors recognized by the targeting ligands, including ligands targeting oocytes and granulosa cells.

### 5.1. Oocyte

The ovarian follicle consists of oocytes encased in somatic cells. It serves as the foundational functional unit of female reproduction [138]. Anovulatory infertility frequently occurs in women with PCOS. High-quality oocytes correlate with effective embryonic development and the likelihood of a successful pregnancy [139]. Follicle development is strictly controlled by pituitary gonadotropins, namely follicle-stimulating hormone (FSH) and *LH*, as well as intraovarian regulatory elements such as steroids, growth factors, and cytokines [140]. PCOS can negatively impact the ovarian microenvironment. Pro-inflammatory cytokines, including *TNFα*, *IL-1β*, and *IL-6*, disrupt oocyte maturation, while a chronic inflammatory state leads to ongoing oxidative stress in the ovaries, causing negative implications for pregnancy. Alterations in the inflammatory microenvironment within the follicular fluid of patients with PCOS result in the activation of inflammatory pathways in granulosa cells, serving as a significant contributor to the adverse symptoms experienced by these patients [141]. Selenium nanoparticles have the potential to mitigate the upregulation of inflammatory cytokines, such as IL-6, TNF-α, and IL-1, thereby offering a therapeutic avenue to alleviate PCOS through their antioxidant and anti-inflammatory effects [135]. Recent studies have explored nanomaterial-based drug delivery systems aimed at enhancing oocyte functionality in relation to fertility [125,142,143]. Researchers have employed active agents to modify the surfaces of nanocarriers, such as chitosan [144]. Chitosan, a natural polysaccharide, finds applications across diverse sectors, including pharmaceuticals and medicine [145,146,147,148]. Chitosan nanoparticles (ChNPs) exhibit significant potential as nanocarriers due to their ability to regulate drug release, minimize dosage and side effects, and enhance drug stability, potency, and bioavailability [149]. Chitosan nanoparticles, utilized as a secure and biocompatible carrier for flaxseed (Linum usitatissimum L.) extract (FSE), demonstrate optimal efficacy in regulating serum concentrations of the analyzed luteinizing hormone and FSH parameters. This improved performance is attributed to the significant role of chitosan nanoparticles in augmenting the stability, bioavailability, transport, and permeability of therapeutically relevant phytochemicals [150]. Bilirubin-grafted chitosan-coated nanoparticles can lower intracellular ROS and Ca^2+^ levels, effectively safeguarding oocyte quality and facilitating oocyte maturation, all while preserving the fertilization capacity and developmental potential of mature oocytes. Furthermore, reports indicate that the mannose binding site may influence the recognition and fertilization capabilities of oocytes, positioning it as a potential vehicle for targeted drug or molecule delivery to oocytes [151].

### 5.2. Granulosa Cells

Granulosa cells, together with oocytes, form the follicle within the ovary [152]. These cells proliferate pre-ovulation and are critical for the healthy development of oocytes and the fertilization process that follows [153]. In PCOS patients, excessive androgens can induce metabolic disturbances in granulosa cells, contributing to ovarian dysfunction [152]. Various copper nanoparticles enhance the functionality of ovarian granulosa cells, modulating the secretion of progesterone, testosterone, and estradiol, offering therapeutic benefits for reproductive disorders [154]. Intriguingly, a study involving bovine ovarian granulosa cells revealed that silver nanoparticles may trigger apoptosis and oxidative stress, consequently impairing ovarian cell function and viability [155].

## 6. Conclusions and Prospect

PCOS represents the most prevalent condition among infertile women. Nanomaterials show potential to alleviate certain therapeutic constraints related to PCOS. They can prolong the half-life of medications, enhance drug bioavailability, and facilitate targeted delivery to specific cells. This approach may extend the therapeutic window, minimize adverse effects, and improve drug efficacy for PCOS treatment. This review consolidates recent advancements in nanomaterial-based drug delivery systems for PCOS management. We categorize and describe the various nanoparticles employed in PCOS therapy, focusing on natural drug nanoparticles and metallic nanoparticles. Additionally, we outline various nanocarriers relevant to PCOS, such as liposomes, carbon nanotubes, quantum dots, and micelles. Finally, we suggest potential targeting ligands for these nanomaterials.

Currently, research into novel drug delivery systems for PCOS is still nascent, indicating significant opportunities for exploration. The application of nanotechnology extends beyond merely enhancing the physicochemical properties of drugs; it also encompasses the improvement of therapeutic efficacy and the reduction of side effects, which is particularly crucial for patients with PCOS who often require long-term management of endocrine and metabolic issues. Nanocrystal drug formulations and nanoemulsion technology represent two promising approaches for nanomedicine delivery [156]. Nanocrystal technology enhances the solubility and dissolution rate of poorly soluble drugs by reducing particle size to the nanoscale, which is vital for certain medications that may be utilized in the treatment of PCOS. For instance, drugs aimed at regulating endocrine function and improving insulin resistance can benefit from the application of nanocrystal technology, leading to increased efficacy and diminished side effects. On the other hand, nanoemulsion technology offers an innovative drug delivery pathway. Nanoemulsions, formed spontaneously from water, oil, surfactants, and co-surfactants, consist of thermodynamically stable, isotropic, transparent, or semi-transparent droplets ranging from 1 to 100 nanometers. This technology can enhance the solubility and bioavailability of drugs, particularly those that are highly hydrophobic. In the context of PCOS treatment, nanoemulsions can serve as effective carriers, protecting drugs from environmental degradation while improving their dispersion and absorption within the body. Notably, there are successful clinical applications of nanomedicine delivery systems. For example, liposomal doxorubicin (Doxil and Myocet) has improved the pharmacokinetics and biodistribution of the drug while reducing cardiac and cutaneous toxicity. Additionally, nab-paclitaxel (Abraxane) has enhanced drug tolerability and bioavailability. These instances illustrate that nanotechnology offers novel strategies for enhancing drug efficacy and patient compliance [156]. Most studies predominantly emphasize the bioavailability and targeting capabilities of nanoparticles, while investigations into nanoparticle toxicity and elimination in the context of PCOS remain scarce, warranting further inquiry. Moreover, comprehensive studies examining the mechanisms and therapeutic impacts of nanoparticles in PCOS are essential for advancing nanomaterial-based drug delivery from laboratory settings to clinical applications. Additionally, further in-depth investigations regarding similar nanoparticle types are needed.

## Figures and Tables

**Figure 1 pharmaceutics-16-01556-f001:**
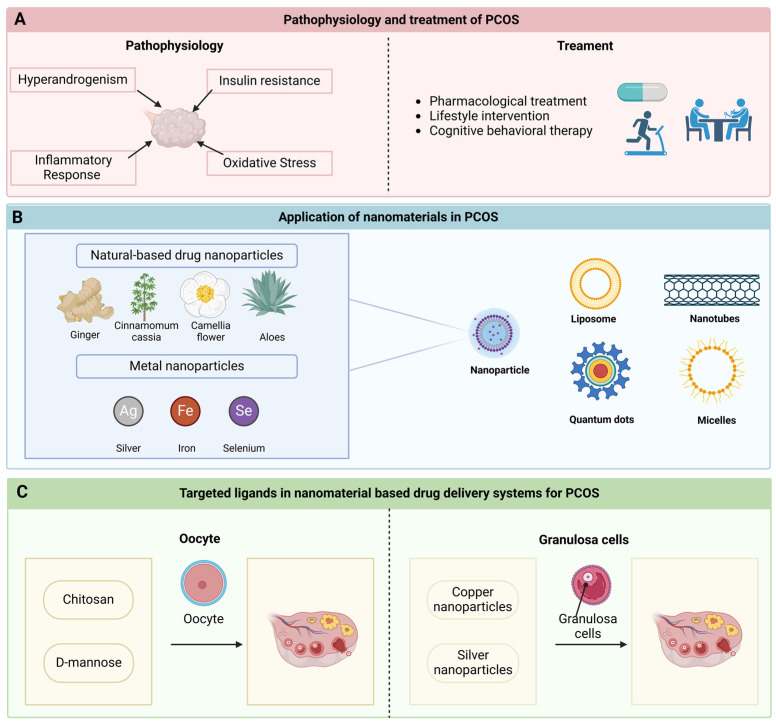
Nanomaterials and targeted ligands in PCOS therapy. (**A**) The pathophysiological mechanisms underlying polycystic ovary syndrome (PCOS) predominantly encompass hyperandrogenism, insulin resistance, inflammatory responses, and oxidative stress. Current therapeutic approaches for PCOS primarily consist of pharmacological interventions, lifestyle modifications, and psychotherapy. (**B**) A diverse array of novel nanomaterials has been employed for the management of PCOS. This includes nanoparticles—natural medicine nanoparticles sourced from ginger, cinnamon, camellia, and aloe vera, as well as metal-based nanoparticles derived from silver, iron, and selenium—along with liposomes, nanotubes, quantum dots, and micelles. (**C**) Oocytes and granulosa cells may serve as targeted ligands, playing a significant role in innovative drug delivery systems. Chitosan and mannose can provide a protective function for the ovaries by binding to oocytes. Moreover, copper and silver nanoparticles possess the capability to transit through granulated cells, thereby influencing ovarian functions.

**Table 1 pharmaceutics-16-01556-t001:** Nanocarriers for PCOS treatment.

Number	Nanocarrier	Key Ingredient	Therapeutic Agent	Research Object	Mechanism	Reference
1	Chitosan nanoparticles	Chitosan	Curcumin	*Rat*	Reduce the levels of serum luteinizing hormone, prolactin, testosterone, and insulin	[59]
2	Ginger nanoparticles	Lipid	Ginger	*Mice*	Elevated the expression of forkhead transcription factor (Foxa2) to mitigate insulin resistance induced by intestinal epithelial cell (IEC) exosomes	[60]
3	Silver nanoparticles	Silver	Cinnamomum cassia	*Rat*	Antioxidant	[61]
4	Silver nanoparticles	Silver	Cinnamomum zeylanicum	*Rat*	Reduce the levels of inflammatory markers such as IL-6, IL-18, and TNF-α	[62]
5	Iron nanoparticles	Iron oxide	Curcumin	*Mice*	Inhibition of ovarian injury cell apoptosis and dehydroepiandrosterone-induced cell apoptosis	[63]
6	Selenium nanoparticles	Chitosan	Selenium dioxide	*Rat*	Reduce androgen synthesis and block the vicious cycle caused by excessive androgen secretion by reducing the expression of androgen receptors	[64]
7	Selenium nanoparticles	Chitosan	Selenium dioxide	*Rat*	Upregulation of PI3K and Akt gene expression reduces insulin sensitivity, lipid profile, sex hormone levels, inflammation, oxidative stress, and mitochondrial functional markers	[65]
8	Liposomes	Glycerol phospholipid	Methoxy derivatives of resveratrol (DMU-212)	*Ovarian granulosa cells*	Increase the secretion of estradiol and progesterone	[66]
9	Carbon nanotubes	Silkworm powder	Nitrogen-doped carbon nanorods (N-CNR)	*Mice*	Reduce fasting blood glucose and improve serum biomarker levels associated with oxidative stress and inflammation	[67]
10	Quantum dot	Polyethylene glycol (PEG)	Metformin	*Hepg2 cells*	Restore glucose uptake and reverse insulin resistance	[68]
11	Micelle	/	Curcumin	*Rat*	Reduced oxidative damage and inflammatory response	[69]

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
