# Peer review of "Polycystic Ovary Syndrome and the Potential for Nanomaterial-Based Drug Delivery in Therapy of This Disease"

_pharmaceutics, 2024, doi:10.3390/pharmaceutics16121556_

Round 1
Reviewer 1 Report
Comments and Suggestions for Authors
The manuscript by D. Qin et al. reviews the recent advances in the developing and optimization a the drug delivery systems based on nanomaterials for treatment of polycystic ovary syndrome, which is dangerous for women fertility. A wide range of natural and synthesized nanoparticles, i.e., natural-based drug nanoparticles, metal nanoparticles, liposomes, carbon nanotubes, quantum dots, micelles are presented in this review and their role carriers for drug delivery transport for PCOS treatment is presented and discussed. An effective route for the optimization of the nanomaterial-based drug delivery systems through targeted ligands modification is also considered. The reference list includes 151 publications, most of them are belonging to the 2020-2023 time period This paper may be interesting for the broad readership including specialists in medicine, pharmacy, nanotechnology and fine organic synthesis.
However, before publication, some issues should be addressed.
1. The referee suggests that the manuscript should be further balanced, so the sections relevant to medicine aspects of PCOS, e.g., its manifestation should be compressed.
2. The reference 48 should be corrected.
3. The relevant review should be cited with an appropriate discussing. https://doi.org/10.1016/j.jddst.2024.106348.
4. The mechanism of nanoparticle action as DDS for PCOS treatment should be more illustrated, especially with targeted ligands modifying.
Reviewer 2 Report
Comments and Suggestions for Authors
The authors have prepared a thorough review of the nature of the condition Polycystic Ovary Syndrome (PCOS) and its current treatment, described the various types of nanomaterials that have been considered as vectors for delivery of therapeutic agents, and offered a short perspective of how nanomaterials may be applied in the treatment of PCOS. The overview of the disease and its current treatment options is detailed and well written, occupying nearly half of the manuscript. The second part of the manuscript talks of nanoparticles - mostly referring to those prepared from natural materials, and then covers liposomes, nanotubes, quantum dots and micelles. Missing here might be a reference to nanocrystalline forms and nanoemulsion formulations of small anti-androgen molecules might be brought in here as a further example of nanomaterial technology being used for enhancing biopharmaceutical properties of antiandrogen drugs which may have off-label use in therapy of PCOS? Finally there is a short section (section 5.1, 5.2) about targeting therapy for PCOS via nanomaterials.
Overall the manuscript title does not accurately reflect the content and might be adapted to something like "Polycystic Ovary Syndrome and the potential for nanomaterial-based drug delivery in therapy of the condition". This title shows that much of the article is about the disease and how it is currently treated plus there is an overview of how nanomaterials could be brought to bear in treating the condition.
Specific points therefore are :
1) Reconsider article title to better reflect content.
2) Add in some discussion of nanocrystalline drug approaches and nanoemulsions.
3) Page 5 line 200 refers to insulin sensitivity and page 5 line 223 notes "insulin sensitisers"- as the most widely used drug adapted from the treatment of type 2 diabetes to the management of symptoms of PCOS is metformin which might be better considered as acting on insulin resistance, I would recommend using this terms through the article. "Insulin sensitisers" usually makes me think of pioglitazone which has only restricted use now in type 2 diabetes and probably has little use in PCOS.
Page6 line 233: it is a rather bold statement to say that nanomedicines "have transformed how we deliver active pharmaceutical compounds". This seems to suggest that large numbers of new medicines are nano. They are not. This sentence might better be "nanomedicines "have significantly enhanced the utility of some medicines", with examples in clinical use today being doxorubicin (liposomes, reduces toxicity), paclitaxel (as albumin nanoparticles, improves tolerability) and aprepitant (nanocrystals, enhances bioavailability). Please check how well references 67 and 68 truly reflect on the clinical utility of nanomedicines. I am not sure they support the claim too well.
Reviewer 3 Report
Comments and Suggestions for Authors
My concern about this review article is given below:
-What are the key mechanisms through which PCOS contributes to anovulatory infertility and metabolic dysfunction, and how might nanomaterial-based treatments intervene in these processes?
-Author should discuss about, How do nanomaterial-based drug delivery systems improve the pharmacokinetics of existing PCOS medications such as metformin and clomiphene citrate?
-What strategies have been employed to ensure targeted delivery to ovarian and endocrine tissues while minimizing systemic side effects in PCOS patients?
-What types of drug release triggers (e.g., pH, temperature, enzyme-sensitive) have been most effective for nanomaterial-based PCOS therapies, and how do they function in the context of ovarian and endocrine tissues?
-How does the incorporation of nanomaterials help mitigate the inflammatory responses often associated with PCOS and potentially improve metabolic and endocrine outcomes?
-What role do targeting ligands, such as peptides or antibodies, play in enhancing the precision of nanomaterial-based drug delivery systems for PCOS treatment?
-What types of drug release triggers (e.g., pH, temperature, enzyme-sensitive) have been most effective for nanomaterial-based PCOS therapies, and how do they function in the context of ovarian and endocrine tissues?
-While the benefits of nanomaterials in prolonging drug half-life and enhancing bioavailability are mentioned, elaborating on the underlying mechanisms through which these improvements occur would strengthen the scientific argument.
-The text mentions that research into nanoparticle toxicity in PCOS is scarce. Including a dedicated section summarizing known nanoparticle toxicity profiles and any relevant safety measures could make the review more comprehension.
-The review mentions natural and metallic nanoparticles. Adding more detail on how these types compare in terms of biocompatibility, potential immune responses, or degradation rates could be beneficial for readers.
-There is a good explanation of the role of oocytes and granulosa cells, but discussing how nanoparticles interact with the unique microenvironment of these cells in PCOS would add to the scientific discussion.
-The benefits of chitosan nanoparticles are noted, but specifying how they can be modified to enhance selectivity for oocytes or granulosa cells, such as through functionalization with specific ligands or conjugation with hormones, would add practical insights.
Comments on the Quality of English Languageno
Round 2
Reviewer 2 Report
Comments and Suggestions for Authors
Authors have considered this reviewer's feedback and used it to further develop their manuscript. It has been improved and reads well and is informative. I have not further comments for the authors to address.